# SELF SPECULATIVE DECODING FOR DIFFUSION LARGE LANGUAGE MODELS

## ABSTRACT

Diffusion-based Large Language Models (dLLMs) have emerged as a competitive alternative to autoregressive models, offering unique advantages through bidirectional attention and parallel generation paradigms. However, the generation results of current parallel decoding methods deviate from stepwise decoding, introducing potential performance degradation, which limits their practical deployment. To address this problem, we propose **S**elf **S**peculative **D**ecoding (SSD), a lossless inference acceleration method that leverages the dLLM itself as both speculative decoding drafter and verifier without auxiliary modules. SSD introduces a self-drafting mechanism where the model generates predictions for multiple positions, then verifies them through hierarchical verification trees in a single forward pass. Unlike traditional speculative decoding that requires separate draft models, SSD eliminates model redundancy and memory overhead by exploiting the dLLM's inherent parallel prediction capability for multiple positions. This self-speculative approach allows the model to progressively verify and accept multiple tokens in a single forward pass. Our experiments demonstrate that SSD achieves up to $3.46\times$ speedup while keeping the output identical to stepwise decoding on open source models such as LLaDA and Dream. Code is provided in the supplementary material and will be made publicly available on GitHub.

## 1 INTRODUCTION

Large Language Models (LLMs) have revolutionized natural language processing, with autoregressive models (ARMs) (Radford et al., 2018; 2019; OpenAI, 2022) dominating the landscape due to their strong performance and straightforward training paradigm (OpenAI, 2023; Touvron et al., 2023). However, the sequential nature of ARMs, where each token is generated conditioned on all previous tokens, inherently limits their inference speed and poses significant challenges for real-time applications. Moreover, this sequential dependency leads to error propagation (Stechly et al., 2023; Valmeekam et al., 2023) and difficulties in maintaining global coherence (Mei et al., 2025).

Recently, diffusion-based Large Language Models (dLLMs) have emerged as a compelling alternative, generating text through iterative denoising of masked segments rather than sequential token prediction (Austin et al., 2021; Gulrajani & Hashimoto, 2024; Nie et al., 2025). This paradigm shift offers several advantages: (1) bidirectional attention mechanisms (Seo et al., 2017) that capture richer contextual dependencies, (2) the potential for parallel token generation within each denoising step, and (3) demonstrated superiority in tasks requiring bidirectional reasoning, such as the reversal curse problem where dLLMs outperform even GPT-4 (Nie et al., 2025). The promise of dLLMs has been further validated by proprietary systems like Mercury Coder (Inception Labs, 2025) and Gemini Diffusion (Google DeepMind, 2025), which achieve inference speeds exceeding 1000 TPS.

Despite these advantages, dLLMs face a challenge in inference efficiency. Unlike ARMs that have native Key-Value (KV) caching support, dLLMs' bidirectional attention mechanism and dynamic token updates across denoising steps prevent straightforward application of traditional KV caching strategies. Recent work has addressed this by proposing adaptive caching frameworks (Liu et al., 2025; Wu et al., 2025), achieving impressive speedups and fundamentally shifting the computational characteristics of dLLMs from compute-bound to memory-bound regimes. As shown in Figure 1, when cache in Fast-dLLM (Wu et al., 2025) is enabled, dLLM generation exhibits considerable memory-bound characteristics at moderate batch sizes ($\leq 8$), where throughput scaling approaches

linear scaling, creating opportunities for speculative decoding approaches (Leviathan et al., 2023; Chen et al., 2023) that can trade computation for reduced latency.

Speculative decoding has proven highly effective for accelerating ARM inference by using a lightweight draft model to propose multiple tokens that are then verified in parallel by the target model (Leviathan et al., 2023; Chen et al., 2023). Recent advances like EAGLE (Li et al., 2024) and Medusa (Cai et al., 2024) have pushed the boundaries of speculative sampling through feature-level autoregression and multiple decoding heads. However, directly applying these techniques to dLLMs is non-trivial due to fundamental differences in their generation paradigms: dLLMs operate on masked positions with iterative unmasking rather than sequential prediction.

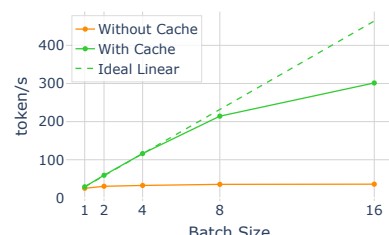

Figure 1: TPS comparison with and without cache showing memory-bound characteristics of LLaDA-8B-Instruct when cache is enabled.

In this paper, we propose **S**elf **S**peculative **D**ecoding (SSD), an acceleration framework that enables dLLMs to act as their own draft models through hierarchical verification. Our key insight is that dLLMs can leverage their parallel prediction capability to generate drafts for multiple positions simultaneously, then verify these drafts hierarchically in a single forward pass. Unlike traditional speculative decoding which requires a separate draft model, SSD eliminates model redundancy and memory overhead. Figure 2 illustrates the key difference between vanilla dLLM inference and our SSD approach, showing how SSD can accept multiple tokens in a single iteration through hierarchical verification, and reduce decoding steps.

Our contributions are threefold: (1) We propose SSD, the first speculative decoding framework for dLLMs that leverages self-drafting to eliminate auxiliary models, enabling dLLMs to generate and verify multiple tokens per iteration through hierarchical verification trees. (2) Extensive experiments on five dLLMs across four benchmarks demonstrate up to 3.46× speedup while maintaining identical generation results. (3) We analyze the acceptance limits of self-drafting and explore strategies to approach this limit, providing insights for future optimizations.

## 2 RELATED WORK

### 2.1 DIFFUSION LANGUAGE MODELS

Diffusion models (Sohl-Dickstein et al., 2015; Ho et al., 2020; Song et al., 2021), originally popularized in image generation (Rombach et al., 2022; Nichol et al., 2022; Saharia et al., 2022), have recently gained attention as an alternative to autoregressive language models for text generation. The expansion from continuous to discrete domains was first studied by Sohl-Dickstein et al. (2015). Subsequently, D3PM (Austin et al., 2021) provided a general framework that models the diffusion forward process as a discrete state Markov chain defined by multiplication of specific transition matrices over discrete time steps. Campbell et al. (2022) later expanded D3PM to continuous time.

The development of diffusion language models has progressed through several stages. Initial approaches like DiffusionBERT (He et al., 2022) and Diffusion-LM (Li et al., 2022) demonstrated the feasibility of applying diffusion to text but were limited to specific tasks or required complex training procedures. DiffuSeq (Gong et al., 2022) extended this to sequence-to-sequence generation, while GENIE (Lin et al., 2023) introduced continuous paragraph denoising for improved text generation.

More recently, research on masked diffusion models (MDMs) derived from the absorbing state diffusion in D3PM has shown promising results both in small-scale models (e.g., MDLM (Sahoo et al., 2024) and RADD (Ou et al., 2025)) and large-scale implementations (e.g., LLaDA (Nie et al., 2025) and Dream (Ye et al., 2025)). LLaDA represents a significant milestone, training an 8B parameter model from scratch that achieves competitive performance with LLaMA3-8B while using 7× fewer training tokens. LLaDA's bidirectional attention and iterative refinement enable it to solve the reversal curse problem, outperforming GPT-4 in reversal reasoning tasks. Extending this line of work, MMaDA (Yang et al., 2025) introduces a class of multimodal large diffusion models featuring a shared probabilistic formulation and a modality-agnostic architecture.

## 2.2 EFFICIENT dLLM INFERENCE

Existing acceleration research on dLLMs focuses on two directions: KV cache optimization and sampling compression. The KV cache direction targets building cache mechanisms for dLLMs due to their bidirectional full attention mechanism, unlike the causal attention of autoregressive models. Works like Block Diffusion (Arriola et al., 2025), Fast-dLLM (Wu et al., 2025) and dLLM-Cache (Liu et al., 2025) explore different caching mechanisms, showing promising performance for speedup. dLLM-Cache employs differentiated caching with long-interval prompt caching and adaptive response caching using a V-verify mechanism that monitors Value vector similarity to selectively update changed tokens. Fast-dLLM introduces both block-wise approximate KV caching and parallel decoding capabilities. These caching optimizations fundamentally shift the computational characteristics of dLLMs from compute-bound to memory-bound regimes.

The sampling compression direction focuses on optimizing the sampling process itself. For the classic low-confidence remasking strategy, several works have introduced novel sampling strategies to dynamically adjust the number of tokens predicted in parallel, thereby improving inference efficiency. Prophet (Yu et al., 2024) leverages early answer convergence, where dLLMs internally identify correct answers before completing all decoding steps, reducing inference steps. Fast-dLLM (Wu et al., 2025) adopts a straightforward approach by selecting tokens with confidence scores exceeding a predefined threshold for parallel decoding. Meanwhile, Ben-Hamu et al. (2025) propose an entropy-bounded (EB) sampler, a drop-in replacement for conventional samplers that leverages an entropy-based unmasking procedure to dynamically decode multiple tokens per step while maintaining a predefined error tolerance. SlowFast Sampling (Wei et al., 2025) dynamically alternates between exploratory and aggressive phases based on generation confidence.

## 2.3 SPECULATIVE DECODING

Speculative decoding accelerates language model inference by having a lightweight draft model propose multiple tokens that are verified in parallel by the target model (Leviathan et al., 2023; Chen et al., 2023). This approach exploits the observation that next-token prediction in autoregressive models underutilizes GPU computational capacity, allowing parallel verification of multiple draft tokens without proportional latency increase.

Traditional speculative decoding uses a separate smaller model as the drafter, but recent advances have explored more sophisticated approaches. EAGLE (Li et al., 2024) performs autoregression at the feature level, reusing top-layer features from the target model to achieve better draft quality. EAGLE-3 (Li et al., 2025) further improves this by abandoning feature prediction in favor of direct token prediction with multi-layer feature fusion. Medusa (Cai et al., 2024) uses multiple decoding heads to generate several candidate continuations simultaneously, while Sequoia (Chen et al., 2024) introduces hardware-aware optimizations for practical deployment.

Recent work has begun incorporating dLLMs into autoregressive speculative decoding. Speculative Diffusion Decoding (Lou et al., 2024) proposes using diffusion models as drafters to accelerate autoregressive model inference, demonstrating that dLLMs can generate high-quality draft sequences for ARMs. However, Using speculative decoding in dLLMs presents unique challenges. The iterative refinement process and masked token prediction of dLLMs require fundamentally different verification strategies than autoregressive models. Our SSD framework addresses these challenges by enabling dLLMs to perform self-speculative decoding, where the model serves as both drafter and verifier without requiring auxiliary models.

## 3 METHODOLOGY

### 3.1 FORMULATION

Diffusion language models generate text through an iterative denoising process. Given a prompt $\mathbf{x}_{\text{prompt}}$ of length $P$ and a target generation length $L$, the model starts with a sequence containing masked tokens:

$$\mathbf{x}^{(0)} = [\mathbf{x}_{\text{prompt}}, \mathbf{m}_1, \mathbf{m}_2, \ldots, \mathbf{m}_L] \qquad (1)$$

where $\mathbf{m}_i$ represents a mask token at position $i$. Through $T$ denoising steps, the model iteratively refines these masked positions to generate the final text.

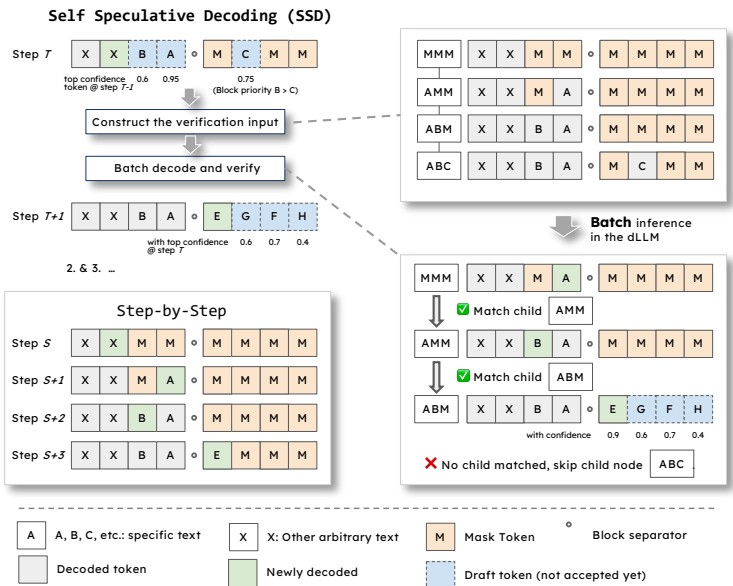

Figure 2: Comparison of stepwise dLLM inference (bottom-left) with our SSD approach. Stepwise inference accepts one token per step following semi-autoregressive block order. SSD leverages self-drafting and hierarchical verification to accept multiple tokens per iteration. When both methods reach the same intermediate state at step $T = S$, SSD generates the next 3 tokens in step $T + 1$ while stepwise requires steps $S + 1$ through $S + 3$ (example with draft length $k = 3$).

At each denoising step $t$, the model predicts tokens for all masked positions simultaneously:

$$p(\mathbf{x}_i^{(t)}|\mathbf{x}^{(t-1)}) = \text{Softmax}(f_\theta(\mathbf{x}^{(t-1)})_i), \quad \forall i \in \mathcal{M}^{(t-1)} \tag{2}$$

where $\mathcal{M}^{(t-1)}$ denotes the set of masked positions at step $t - 1$, and $f_\theta$ is the transformer model with parameters $\theta$.

Unlike autoregressive models that generate tokens sequentially, dLLMs leverage bidirectional attention, allowing each position to attend to all other positions (both before and after). This enables richer contextual understanding.

Diffusion language models like LLaDA (Nie et al., 2025) employ a semi-autoregressive generation strategy where tokens are generated in sequential blocks. The generation positions are partitioned into blocks of size $B$:

$$\mathcal{B}_j = \{P + jB + 1, P + jB + 2, \ldots, P + \min((j+1)B, L)\} \tag{3}$$

where $j \in \{0, 1, \ldots, \lceil L/B \rceil - 1\}$ indexes the blocks.

In this semi-autoregressive framework, tokens in block $\mathcal{B}_j$ must be fully generated before any tokens in block $\mathcal{B}_{j+1}$ are considered for acceptance, regardless of their confidence scores. Even if tokens in later blocks exhibit higher confidence than those in the current block, they cannot be accepted until the current block is complete.

### 3.2 SELF SPECULATIVE DECODING

#### 3.2.1 OVERVIEW

SSD accelerates dLLM inference by enabling the model to act as its own speculative decoder. The key insight is that dLLMs can generate high-quality drafts for all positions in parallel, then verify these drafts through hierarchical tree structures where parent nodes' verified tokens progressively improve child nodes' predictions. As illustrated in Figure 2, SSD transforms the vanilla step-by-step denoising process into multi-token predicting framework per iteration.

### 3.2.2 SELF-DRAFTING AND SELECTION

The self-drafting mechanism is the foundation of SSD. At the beginning of inference and after each hierarchical verification round, the dLLM generates draft tokens for all remaining masked positions in a single forward pass:

$$\mathbf{d}^{(t)}, \mathbf{c}^{(t)} = \text{SelfDraft}(\mathbf{x}^{(t)}, f_\theta) \tag{4}$$

where $\mathbf{d}^{(t)} \in \mathbb{N}^{|\mathcal{M}^{(t)}|}$ contains draft tokens for masked positions $\mathcal{M}^{(t)}$, and $\mathbf{c}^{(t)} \in [0,1]^{|\mathcal{M}^{(t)}|}$ contains confidence scores computed from the softmax probabilities.

From these drafts, we employ a **greedy** selection strategy to choose up to $N$ positions for verification while respecting the semi-autoregressive constraints. Following the block-wise generation order, only positions from block $\mathcal{B}_j$ are considered if any position in $\mathcal{B}_j$ remains ungenerated. Within the current block, positions with higher confidence scores are selected first. Only when the current block has fewer than $N$ remaining positions do we select additional positions from block $\mathcal{B}_{j+1}$. This self-drafting approach eliminates the need for auxiliary models while exploiting dLLMs' inherent parallelism, ensuring that earlier blocks are fully completed before later blocks begin generation.

### 3.2.3 HIERARCHICAL VERIFICATION TREE

Given selected candidates $\mathcal{C}$, SSD constructs a hierarchical verification tree where a child node is only valid when its parent node's token exactly matches the step-wise generation result. The tree is rooted at the current state with already confirmed tokens, and each node represents a state with a specific draft token placed at a candidate position. We adopt a **greedy strategy** that builds a linear chain where each node has at most one child, verifying candidates in priority order with exactly $N + 1$ nodes for draft length $N$.

### 3.2.4 BATCH VERIFICATION

The hierarchical verification is the core of SSD's acceleration. All nodes in the verification tree are processed in a single batch forward pass:

$$\mathbf{P}_{\text{all}} = f_\theta([\mathbf{x}_{\text{node}_1}, \mathbf{x}_{\text{node}_2}, \dots, \mathbf{x}_{\text{node}_n}]) \tag{5}$$

Each node's sequence contains its corresponding draft tokens placed at candidate positions. The tree traversal follows a hierarchical validation: child nodes can only be accepted if their parent nodes' predictions exactly match the expected tokens. Starting from the root, we verify whether the model's prediction matches the expected token for each child node, and continue traversal only for validated branches.

During batch verification, we compare each parent node's prediction with its child node's draft token. If a match is found, we proceed to verify that child node; otherwise, we terminate the verification for this iteration. All accepted tokens are incorporated into the decoded sequence, and drafts are updated for the next iteration without duplicate context.

The complete SSD algorithm is presented in Algorithm 1 (Appendix A). Notably, our **greedy** verification strategy with draft length $N$ produces exactly $N + 1$ verification nodes in the hierarchical tree. In the best case where all $N$ candidate tokens pass verification, SSD accepts $N + 1$ tokens in a single iteration—the $N$ verified drafts plus one token from the leaf node's prediction.

## 4 EXPERIMENT

### 4.1 EXPERIMENT SETUP

In this section, we present the evaluation setup, including the models, tasks, metrics, hyperparameters, and comparison baselines.

**Models.** We evaluate two mainstream families of dLLMs: LLaDA (LLaDA-Base, LLaDA-8B-Instruct (Nie et al., 2025), and LLaDA-1.5 (Zhu et al., 2025)) and Dream (Dream-7B-Base and Dream-7B-Instruct (Ye et al., 2025)). The LLaDA family spans both base and instruction-tuned variants, while the Dream family focuses on medium-scale general-purpose models.

**Datasets.** We consider four widely used benchmarks: GSM8K (Cobbe et al., 2021), MATH (Hendrycks et al., 2021), HumanEval (Chen et al., 2021), and MBPP (Austin et al., 2021), covering both mathematical reasoning and code generation. All benchmarks are evaluated under the zero-shot setting.. To ensure reproducibility, we rely on the standardized `lm-eval` (Gao et al., 2024) library. For efficiency, we further subsample 10% of the GSM8K and MATH datasets for evaluation by setting `limit` parameter in `lm-eval` to 0.1.

**Metrics.** All experiments are conducted on a single NVIDIA A100 80GB GPU. The primary evaluation metric is throughput, defined as the number of valid tokens generated per second (TPS), which directly reflects the efficiency of dLLMs under different decoding strategies. Valid tokens include all tokens up to but not including the `<eos>` token. We also report the number of decoding steps consumed during generation. Given that model inference is typically memory-bound, the ideal speedup ratio can be estimated based on the number of decoding steps.

**Baseline.** We adopt vanilla semi-autoregressive decoding with cache as the baseline method. For verification tree decoding, we use greedy search as the default strategy. For cache management, we employ the dual-cache mechanism from Fast-dllm (Wu et al., 2025).

**Hyperparameters.** Cache and self-speculative decoding involve three key hyperparameters: block length, draft length, and cache refresh interval. Unless otherwise specified, we set block length = 8, cache refresh interval = 8, and draft length $\in \{3, 4, 5\}$. The cache refresh interval specifies that the key–value states cached for all positions are refreshed once every fixed number of decoded tokens. We set generation length for all dLLMs to 256 tokens.

## 4.2 MAIN RESULTS

We report our result of five dLLMs on four benchmarks mentioned above. Since our method is theoretically lossless, we omit the accuracy numbers here.

Table 1: Comparison of SSD speed across different models and benchmarks. Configurations achieving the highest acceleration are highlighted in bold.

| Model | Method | Draft Length | GSM8K TPS | Step↓ | MATH TPS | Step↓ | HumanEval TPS | Step↓ | MBPP TPS | Step↓ | Mean TPS | Step↓ |
|---|---|---|---|---|---|---|---|---|---|---|---|---|
| LLaDA Base | Origin | - | 14.99 | - | 26.34 | - | 8.30 | - | 5.74 | - | 13.84 | - |
| | Origin+SSD | 3 | 28.73 (1.92×) | 52.9% | 52.35 (1.99×) | 55.1% | **18.28** (2.20×) | 57.3% | 10.17 (1.77×) | 56.7% | 27.38 (1.98×) | 55.5% |
| | Origin+SSD | 4 | **30.47** (2.03×) | 55.6% | **53.61** (2.04×) | 58.2% | 18.01 (2.17×) | 60.4% | **10.31** (1.78×) | 60.0% | **28.10** (2.03×) | 58.6% |
| | Origin+SSD | 5 | 30.39 (2.03×) | 56.8% | 52.54 (1.99×) | 59.7% | 17.56 (2.12×) | 61.6% | 9.86 (1.72×) | 61.5% | 27.59 (1.99×) | 59.9% |
| LLaDA Instruct | Origin | - | 26.91 | - | 27.34 | - | 26.13 | - | 16.98 | - | 24.34 | - |
| | Origin+SSD | 3 | 57.04 (2.12×) | 58.1% | 56.52 (2.07×) | 54.0% | 53.28 (2.04×) | 57.7% | 30.21 (1.78×) | 59.8% | 49.26 (2.02×) | 57.4% |
| | Origin+SSD | 4 | 60.07 (2.23×) | 61.3% | **58.98** (2.16×) | 57.2% | **55.28** (2.12×) | 60.7% | **31.15** (1.83×) | 63.0% | **51.37** (2.11×) | 60.6% |
| | Origin+SSD | 5 | **60.39** (2.24×) | 63.2% | 57.99 (2.12×) | 58.9% | 54.14 (2.11×) | 62.5% | 30.33 (1.79×) | 65.2% | 50.71 (2.08×) | 62.5% |
| LLaDA 1.5 | Origin | - | 26.50 | - | 27.66 | - | 27.25 | - | 18.58 | - | 25.00 | - |
| | Origin+SSD | 3 | 57.48 (2.17×) | 58.0% | 57.26 (2.07×) | 56.1% | 55.51 (2.04×) | 57.7% | 33.32 (1.79×) | 59.1% | 50.89 (2.04×) | 57.7% |
| | Origin+SSD | 4 | **61.43** (2.32×) | 61.1% | **59.01** (2.13×) | 59.1% | **58.18** (2.13×) | 60.7% | **33.68** (1.81×) | 62.3% | **53.08** (2.12×) | 60.8% |
| | Origin+SSD | 5 | 60.72 (2.29×) | 63.1% | 58.29 (2.11×) | 60.8% | 57.41 (2.11×) | 62.4% | 32.84 (1.77×) | 64.6% | 52.32 (2.09×) | 62.7% |
| Dream Base | Origin | - | 13.29 | - | 26.94 | - | 12.62 | - | 5.94 | - | 14.70 | - |
| | Origin+SSD | 3 | 28.78 (2.17×) | 52.8% | 54.42 (2.02×) | 55.2% | 21.66 (1.72×) | 53.3% | 14.03 (2.36×) | 61.4% | 29.72 (2.02×) | 55.7% |
| | Origin+SSD | 4 | 30.16 (2.27×) | 55.4% | 57.16 (2.12×) | 58.1% | 19.28 (1.53×) | 56.3% | **15.02** (2.53×) | 64.7% | 30.41 (2.07×) | 58.6% |
| | Origin+SSD | 5 | **30.25** (2.28×) | 56.6% | **58.65** (2.18×) | 59.8% | **22.54** (1.79×) | 57.7% | 14.70 (2.47×) | 66.5% | **31.54** (2.15×) | 60.2% |
| Dream Instruct | Origin | - | 14.43 | - | 26.73 | - | 17.49 | - | 6.37 | - | 16.26 | - |
| | Origin+SSD | 3 | 32.80 (2.27×) | 60.7% | 56.33 (2.11×) | 56.6% | 36.49 (2.09×) | 59.8% | 18.61 (2.92×) | 70.3% | 36.06 (2.22×) | 61.9% |
| | Origin+SSD | 4 | 35.13 (2.43×) | 64.2% | 58.47 (2.19×) | 59.6% | 39.18 (2.24×) | 63.4% | 21.20 (3.33×) | 74.6% | 38.50 (2.37×) | 65.5% |
| | Origin+SSD | 5 | **36.63** (2.54×) | 66.4% | **60.02** (2.25×) | 61.4% | **39.61** (2.26×) | 65.3% | **22.07** (3.46×) | 77.4% | **39.58** (2.43×) | 67.6% |

Table 1 reports the performance of LLaDA and Dream families under different caching strategies across four representative reasoning and coding benchmarks (GSM8K, MATH, HumanEval, MBPP). We evaluate models in terms of throughput speedup (TPS) and step reduction ratio (Step↓), where higher values indicate improved inference efficiency.

**Effectiveness of SSD.** Our results demonstrate that SSD caching is a universally effective strategy for accelerating inference. Across all models and benchmarks, employing SSD caching more than doubles the mean throughput (TPS) while reducing computational steps by over 50%. The performance gains are most pronounced for Dream-Instruct, which achieves a **3.46×** mean speedup (from 6.37 to 22.07 TPS) and a **77.4%** reduction in decoding steps with a draft length of 5. Even for other

models like LLaDA-Instruct, the speedup is substantial, reaching $2.11\times$. This confirms that SSD provides a robust and significant efficiency boost irrespective of model architecture.

**Comparison across model families.** A direct comparison reveals that the Dream model family benefits more from SSD than the LLaDA family. For instance, at a draft length of 4, Dream-Instruct achieves a mean speedup of $\mathbf{2.37\times}$, which is notably higher than the $\mathbf{2.11\times}$ speedup observed for LLaDA-Instruct. This trend holds across model variants, suggesting that the architectural properties of Dream models may allow for more effective exploitation of cached intermediate representations.

**Impact of draft length.** The optimal draft length is not universal, but depends on the specific model and benchmarks. For the **LLaDA family**, performance peaks at a draft length of 4, with a slight regression observed at a length of 5. For example, LLaDA-1.5's mean TPS drops from 53.08 to 52.32. In contrast, models in the Dream family consistently benefit from a longer draft length, with both mean TPS and step reduction improving from length 4 to 5. This suggests that while a moderate draft length is optimal for LLaDA, the Dream architecture can effectively leverage longer speculative sequences to further boost efficiency.

**Overall efficiency improvements.** In summary, our experiments robustly validate SSD as a model-agnostic accelerator, consistently delivering an approximate $2.0\times$–$3.46\times$ speedup by reducing decoding steps by approximately 50–70%. The varying benefits across model families and draft lengths underscore the interplay between model architecture and caching strategy, with the Dream architecture showing particular promise for caching-based acceleration.

### 4.3 INFLUENCE OF SEQUENCE LENGTH ON ACCELERATION.

In this section, we investigate the effect of sequence length variation on model acceleration. We adopt the same subset of GSM8K and randomly sample 10% of MBPP following the procedure described in the experimental setup. Unless otherwise specified, all configurations remain consistent with the default settings. All experiments are conducted with a fixed draft length of 3.

To disentangle the contribution of different components of the sequence, we consider two factors: the number of prefill tokens and the generation length. For prefill tokens, we vary the number of few-shot examples from 0 to 3. For generation length, we vary the output length across 128, 256, and 512 tokens. We then compare the baseline and our proposed method using tokens-per-second (TPS) as the evaluation metric.

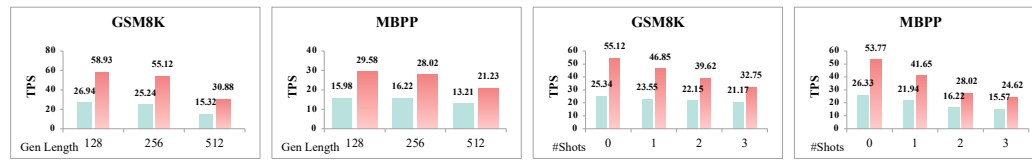

Figure 3: Impact of sequence length on SSD acceleration with LLaDA-8B-Instruct. (Left) Effect of generation length (128-512 tokens) on throughput for GSM8K and MBPP. (Right) Effect of prompt length via few-shot examples (0-3 shots) on throughput for GSM8K and MBPP.

We begin by analyzing the impact of generation length on throughput. Across both GSM8k and MBPP, TPS decreases steadily as the number of output tokens grows from 128 to 512, reflecting the cumulative cost of token-by-token decoding. On GSM8K with cache-only decoding, TPS drops from 26.94 at 128 tokens to 15.32 at 512 tokens, while MBPP shows a similar decline (15.98 $\rightarrow$ 13.21). Importantly, the SSD strategy consistently provides acceleration across all settings: at shorter sequences the benefit is most pronounced (58.93 vs. 26.94 on GSM8K at 128 tokens), while at longer sequences the gain remains significant (30.88 vs. 15.32 at 512 tokens). These results demonstrate that our method is effective across different generation lengths, with particularly strong benefits at moderate lengths that dominate many real-world applications.

We next investigate the influence of few-shot exemplars in the prompt. Increasing the number of shots consistently reduces throughput due to larger prefill overhead. On GSM8k with cache-only decoding, TPS decreases from 25.34 at zero-shot to 21.17 at three-shot, and MBPP exhibits a similar

pattern (26.33 → 15.57). The SSD method alleviates this cost, but the relative speedup is not uniform. At zero-shot, the acceleration is nearly 2.4× (61.24 vs. 25.34 on GSM8k), while at three-shot the gain narrows to about 1.5× (32.75 vs. 21.17). A similar trend holds for MBPP. This indicates that our method is particularly effective when the prefill is short, where memory reuse dominates, while its advantage becomes less dramatic as prompt length grows and prefill computation increasingly dominates runtime. Nevertheless, even under long-prompt conditions, SSD provides meaningful efficiency gains, highlighting its robustness across diverse prompting scenarios

## 5 DISCUSSION

### 5.1 ACCEPTANCE RATE LIMITS

For speculative decoding in AR models, the quality of draft generation largely determines the upper bound of acceleration. Since we use self-draft, for a step-by-step generated token to be potentially accepted through hierarchical verification, it must appear among the top candidate tokens at the corresponding position during draft generation. Our greedy strategy selects only the top-1 confidence token for each position in the draft. This implies an inherent acceptance rate limit for self-draft.

Our greedy hierarchical verification strategy constructs a verification sequence batch of size $N + 1$ for draft length $N$, where the root node represents the initial state with no verified draft tokens and the leaf node represents the final state with all $N$ draft tokens verified and accepted. Complete verification succeeds when each position's top candidate token in the draft, sorted by confidence, matches the step-by-step generation result exactly. Otherwise, verification fails at some tree node, rejecting all subsequent tokens from that point. Figure 4(a) illustrates a common failure case where the draft contains the correct token IDs but in a different generation order than step-by-step decoding.

Under the greedy strategy, the verification tree has each non-leaf node with only one child, where the top candidate token at the next highest-confidence position is accepted as expected. Theoretically, expanding the verification tree with more nodes to accommodate different token generation orders in the draft and non-top-1 tokens accepted by step-wise decoding could match more possibilities.

To approximate the limit acceptance rate and potential reduction in model forward passes, we make the following setting: we record the logits and corresponding top token candidates from each round, where each round consists of draft length $N + 1$ steps (In SSD, each step will at most accept $N + 1$ tokens). During step-wise generation, we check how many tokens are matched by the draft's top-$k$ candidates. Only matched tokens are considered as potentially accepted by SSD's hierarchical verification without additional steps.

Table 2: Step (forward pass) reduction ratio in forward passes under different draft lengths $N$ and top-$k$ candidate selection on MATH using LLaDA-8B-Instruct.

| Draft Length | $k = 1$ | $k = 2$ | $k = 3$ | $k = 4$ | $k = 5$ | Upper Bound |
|:---:|:---:|:---:|:---:|:---:|:---:|:---:|
| 3 | 68.4% | 73.4%(+5.0%) | 73.8%(+0.4%) | 74.6%(+0.8%) | 74.6%(+0.0%) | 75.0% |
| 4 | 72.3% | 77.7%(+5.4%) | 78.1%(+0.4%) | 78.5%(+0.4%) | 78.9%(+0.4%) | 80.0% |
| 5 | 71.5% | 79.3%(+7.8%) | 80.1%(+0.8%) | 80.9%(+0.8%) | 82.0%(+1.1%) | 83.3% |

Table 2 presents the reduction ratio in forward passes for generation length 256. The Upper Bound column represents the theoretical limit when all draft tokens are perfectly accepted, which equals $N/(N+1)$ for draft length $N$. The percentages show the actual reduction achieved when considering the draft's top-$k$ confidence candidates at each position for potential acceptance, compared to vanilla step-by-step decoding.

As shown in Table 2, higher $k$ values provide relatively minor improvements to the acceptance rate limit when considering non-top-1 confidence tokens decoded step-by-step. Specifically, we observe that the primary gain comes from $k = 1$ to $k = 2$ with 5.0%-7.8% improvement, while further increasing $k$ yields negligible benefits.

Consider constructing a complete $k$-ary verification tree ($k > 1$) where we explore all $k$ top candidates at each position. The total number of nodes in such a tree would be:

$$\text{Tree Size} = \sum_{i=0}^{N} k^i = \Theta(k^N), \quad k > 1 \tag{6}$$

This exponential growth from the greedy strategy's $\Theta(N)$ nodes means that even for $k = 2$, the verification batch size would grow from $N + 1$ to $2^{(N+1)} - 1$ nodes. Although $k = 2$ provides 5.0%-7.8% additional forward reduction compared to $k = 1$, the exponentially larger verification batch would push the inference into the memory-bound regime, where throughput saturates and negates the benefits from reduced forward passes.

## 5.2 Trade-off between Acceptance Rate and Verification Tree Size

While $k$-ary verification trees with $k > 1$ could theoretically improve acceptance rates, their $\Theta(k^N)$ growth pushes inference out of the memory-bound regime. Through case studies, we found out-of-order generation is the most common failure case for the greedy strategy. Specifically, even when all step-by-step tokens are among the draft's top-1 candidates, dLLM generation exhibits out-of-order behavior as shown in Figure 4(a), where candidates match but generation order differs. To address this, we augment the greedy strategy's $N + 1$ nodes with additional verification nodes: each non-leaf node can verify its grandchild when out-of-order occurs, as illustrated in Figure 4(b), yielding $2N - 1$ total nodes. Critically, we do not add children after branching nodes, which allows accepting one additional token for out-of-order cases without exponential growth.

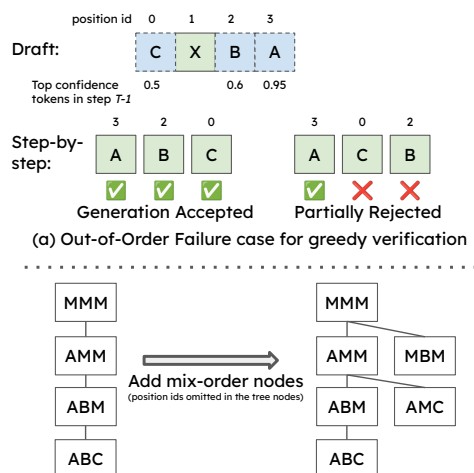

(a) Out-of-Order Failure case for greedy verification

(b) Greedy with mix-order verification tree

Figure 4: (a) Decoding may exhibit out-of-order generation. (b) Mix-order strategy supplements acceptance possibilities under out-of-order conditions.

Table 3 shows that while mix-order strategy achieves 2.5%-3.1% additional step reduction, it requires 50%-67% larger verification batches (e.g., from 5 to 8 nodes for draft length 4). This modest improvement does not justify the increased batch size, so we still adopt the greedy strategy for best SSD acceleration. This demonstrates that efficiently balancing verification tree size and acceptance rate remains worthy of further investigation.

Table 3: Reduction in forward passes with mix-order strategy on MATH using LLaDA-8B-Instruct.

| Draft Length | Greedy | | Greedy + Mix Order | | Upper Bound |
|---|---|---|---|---|---|
| | Verification Batch Size | Step Reduction Ratio | Verification Batch Size | Step Reduction Ratio | |
| 3 | 4 | 59.0% | 6(+2) | 61.5%(+2.5%) | 75.0% |
| 4 | 5 | 62.1% | 8(+3) | 65.2%(+3.1%) | 80.0% |
| 5 | 6 | 63.3% | 10(+4) | 66.4%(+3.1%) | 83.3% |

## 6 Conclusion

We presented SSD (Self Speculative Decoding), enabling diffusion language models to act as their own speculative decoders through parallel self-drafting and hierarchical verification. Our experiments demonstrate that SSD achieves up to 3.46× speedup while maintaining identical generation, making dLLMs more practical for real-world deployment. Future work includes exploring adaptive verification trees and extending the self-speculative principle to other non-AR generation paradigms.

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

## A    ALGORITHM DETAILS

In this section, we provide the detailed algorithm for Self Speculative Decoding (SSD). Algorithm 1 presents the complete procedure, including self-drafting, hierarchical tree construction, and batch verification steps.

---

**Algorithm 1** SSD: Self Speculative Decoding

---

1: **Input:** Prompt $\mathbf{x}_{\text{prompt}}$, model $f_\theta$, generation length $L$, draft length $k$
2: **Output:** Generated sequence
3: Initialize $\mathbf{x}$ with prompt and $L$ mask tokens
4: $\mathbf{d}, \mathbf{c} \leftarrow \text{SelfDraft}(\mathbf{x}, f_\theta)$ {Model drafts for itself}
5: confirmed $\leftarrow \emptyset$
6: **while** masked positions exist **do**
7:    $\mathcal{C} \leftarrow \text{SelectDraftCandidates}(\mathbf{x}, \mathbf{d}, \mathbf{c}, k)$
8:    **if** $|\mathcal{C}| < k$ **then**
9:       Fallback to standard decoding for remaining positions
10:    **else**
11:       tree $\leftarrow \text{BuildHierarchicalTree}(\mathcal{C}, \text{confirmed})$
12:       new_confirmed $\leftarrow \text{HierarchicalVerify}(\text{tree}, f_\theta)$
13:       Update $\mathbf{x}$ with new_confirmed tokens {Self-speculative effect}
14:       $\mathbf{d}, \mathbf{c} \leftarrow \text{UpdateDrafts}(\text{new\_confirmed}, f_\theta)$
15:    **end if**
16: **end while**
17: **return** $\mathbf{x}[P :]$ {Return generated text}

---

The algorithm demonstrates several key aspects of SSD:

- **Self-drafting:** The model generates draft tokens for all positions simultaneously (line 4).

- **Greedy verification:** For draft length $N$, the algorithm produces $N + 1$ verification nodes in the hierarchical tree, and at most $N + 1$ token will be decode in a single step.

