# OpenReview forum: "Self Speculative Decoding for Diffusion Large Language Model"
_ICLR.cc/2026/Conference — ICLR 2026 Conference Withdrawn Submission_

### Official Review · Reviewer_aNdz · 2025-10-17

**Soundness:** 2
**Presentation:** 2
**Contribution:** 1
**Rating:** 2
**Confidence:** 5

**Summary:**

This paper introduces Self-Speculative Decoding (SSD), a lossless inference acceleration method specifically designed for Diffusion-based Large Language Models (dLLMs). The core problem addressed is that while dLLMs offer advantages like bidirectional attention, their parallel decoding methods can deviate from standard stepwise decoding, leading to performance degradation. SSD solves this by leveraging the dLLM itself to act as both the "drafter" and "verifier" in a speculative decoding framework. The comprehensive experiments show that SSD can achieve impressive acceleration compared with vanilla inference paradigm.

**Strengths:**

A crucial advantage is that the method is lossless, meaning the generated text is identical to that of the baseline stepwise decoding. This guarantees that the acceleration does not come at the cost of performance or output quality.

**Weaknesses:**

The method proposed in this paper appears rather trivial, as it is almost entirely transferred from speculative decoding (SD) in autoregressive models (ARMs), without introducing substantial innovation. The main challenge of implementing SD in diffusion large language models (dLLMs) lies in the fact that, unlike ARMs, dLLMs employ bidirectional attention and are compute-bound during inference. When batch inference is used, it often degenerates into sequential verification.

The authors attempt to mitigate this issue by leveraging the *DualCache* mechanism from Fast-dLLMs[1], but the improvement remains limited. As shown in Figure 3, the authors discuss the acceleration results on GSM8K with 0–3 shots, where the speedup effect is shown to diminish progressively. However, typical GSM8K evaluations usually adopt the 4-shot setting [1][2]. Therefore, I believe that SSD is not suitable for typical usage scenarios and still fails to address the fundamental challenges of applying SD in dLLMs.

[1] Wu C, Zhang H, Xue S, et al. Fast-dllm: Training-free acceleration of diffusion llm by enabling kv cache and parallel decoding. arXiv preprint arXiv:2505.22618, 2025.
[2] Nie S, Zhu F, You Z, et al. Large language diffusion models. arXiv preprint arXiv:2502.09992, 2025.

**Questions:**

1. To the best of my knowledge, vanilla speculative decoding involves distribution verification. Why did the authors not extend this mechanism to SSD, instead of adopting a greedy strategy?

2. As both methods aim to accelerate decoding, the authors should compare the speed of their approach with the parallel decoding proposed in [1] in regular few-shot settings. Alternatively, could SSD and parallel decoding be combined to further improve efficiency?

[1] Wu C, Zhang H, Xue S, et al. Fast-dllm: Training-free acceleration of diffusion llm by enabling kv cache and parallel decoding. arXiv preprint arXiv:2505.22618, 2025.

---

### Official Review · Reviewer_2SH6 · 2025-10-24

**Soundness:** 1
**Presentation:** 2
**Contribution:** 1
**Rating:** 2
**Confidence:** 4

**Summary:**

This paper proposes Self-Speculative Decoding (SSD), an efficient and theoretically lossless inference framework tailored for diffusion-based large language models (dLLMs). Unlike prior speculative decoding methods that rely on auxiliary draft models, SSD allows a dLLM to generate and verify its own token drafts within a single forward pass. By integrating self-drafting and hierarchical verification, SSD bridges the gap between parallel and stepwise decoding, ensuring identical outputs while substantially reducing decoding steps. Experimental results across multiple dLLMs demonstrate that SSD delivers up to 3.46× speedup without compromising generation quality.

**Strengths:**

The proposed Self Speculative Decoding (SSD) method enables theoretically lossless generation and significantly reduces the number of decoding steps (forward passes) by allowing diffusion LLMs to self-draft and verify multiple tokens in a single pass, achieving up to 3.46× speedup while maintaining outputs identical to stepwise decoding, a balance of efficiency and fidelity.

**Weaknesses:**

1. In the experiments, the setting of generation length = 256 and block length = 8 was used, which is a common setting. This is because a smaller block length of dual cache leads to less computation during forward passes, marginally alleviating the compute bound problem of batch inference of dLLM. The authors should provide more experiments on common block sizes, e.g., 32 to demonstrate the generability of SSD.

2. This method is essentially a straightforward transfer of self speculative decoding from autoregressive models (ARMs) to dLLMs, with no substantive analysis or use of properties that are specific to dLLMs relative to ARMs.

3. The paper claims that all benchmarks are evaluated under the zero-shot setting, but this results in poor response performance. However, adding more few-shot examples, as discussed in Figure 3, leads to a decrease in speedup. Thus, the acceleration effect of SSD under the same commonly used few-shot setting is relatively limited.

**Questions:**

1. Have the authors encountered situations where, during verification, the model’s batch inference shows some discrepancy compared with single-instance inference [1]

2. The authors mitgrate compute bound of dLLMs by applying DualCache proposed in Fast-dLLMs [2]. However, as shown in the original paper of Fast-dLLMs, DualCache could be lossy in some cases. Therefore, the lossless claimed in this paper is compared with a weak baseline.

3. Table 2 shows the Step (forward pass) reduction ratio in forward passes under different draft lengths N and top-k candidate selection. What is the token per second in each case?

4. Did the authors compare the experimental results with other common training-free decoding strategies and dLLM vanilla decoding without using dual cache?

[1] Agrawal, Sudhanshu, et al. "Spiffy: Multiplying Diffusion LLM Acceleration via Lossless Speculative Decoding." arXiv preprint arXiv:2509.18085 (2025).
[2] Wu, Chengyue, et al. "Fast-dllm: Training-free acceleration of diffusion llm by enabling kv cache and parallel decoding." arXiv preprint arXiv:2505.22618 (2025).

---

### Official Review · Reviewer_LV6E · 2025-10-27

**Soundness:** 3
**Presentation:** 3
**Contribution:** 3
**Rating:** 6
**Confidence:** 5

**Summary:**

The paper introduces **Self-Speculative Decoding (SSD)**, a novel inference acceleration framework for diffusion-based large language models (dLLMs). Unlike conventional speculative decoding—which requires a separate lightweight draft model—SSD enables the dLLM itself to serve as both the drafter and verifier. It achieves this by leveraging the model’s inherent parallel prediction capability to produce self-drafts for multiple masked positions, followed by hierarchical batch verification within a single forward pass. This “self-speculative” approach eliminates redundancy and memory overhead from auxiliary models while maintaining identical generation results compared to stepwise decoding.

Empirically, SSD achieves up to 3.46× speedup and 50–70% step reduction across several open-source dLLMs (e.g., LLaDA, Dream) on reasoning and code benchmarks such as GSM8K, MATH, HumanEval, and MBPP. The method is lossless—it produces outputs identical to baseline decoding—and demonstrates consistent performance gains across architectures and draft lengths. The paper further provides an analysis of the theoretical acceptance rate limits, explores the trade-off between verification tree size and efficiency, and discusses extensions such as mix-order verification strategies.

**Strengths:**

1.  The paper presents a novel self-speculative decoding framework that eliminates the need for an auxiliary draft model in speculative decoding. By enabling a diffusion-based LLM to act as both drafter and verifier, the authors introduce a new paradigm of self-drafting and hierarchical verification, extending speculative decoding beyond autoregressive models
2. The methodology is well-motivated and well formulated, with clear algorithmic steps and theoretical analysis of acceptance-rate limits and verification-tree complexity.
3. SSD provides significant speedups compared to original dLLM baseline. Experiments span multiple dLLM architectures (LLaDA, Dream) and four standard reasoning and code benchmarks, demonstrating consistent 2–3.5× speedups without loss of generation fidelity.

**Weaknesses:**

1. While the experiments cover a diverse set of reasoning and code-generation datasets (GSM8K, MATH, HumanEval, MBPP), they primarily involve short to medium-length reasoning chains. More challenging long-form reasoning benchmarks such as AIME, MATH-500, or GPQA-Diamond could better reveal SSD’s limitations, as longer logical dependencies may reduce the acceptance rate of self-drafted tokens and affect the effective speedup. Evaluating SSD on such tasks would clarify its robustness under deeper reasoning scenarios.

2. The evaluation reports throughput (TPS) and step reduction but omits fine-grained latency and memory analyses, especially across different batch sizes. Since SSD introduces additional verification overhead and batch expansion per step, it remains unclear how well SSD scales in wall-clock latency or memory efficiency compared to other baselines applying speculative decoding for vLLM.

3. Although SSD demonstrates consistent gains on LLaDA and Dream, both are diffusion-based models with similar semi-autoregressive designs. The method’s applicability to non-diffusion or hybrid paradigms (e.g., masked AR models or multimodal diffusion models like MMaDA) remains speculative.

**Questions:**

1. How does SSD perform on longer reasoning benchmarks (e.g., AIME or GPQA) where token dependencies span many denoising steps? Does the acceptance rate of self-drafted tokens degrade significantly with deeper reasoning chains?

2. Could the authors provide end-to-end latency and memory breakdowns (not just throughput) to quantify the real-world efficiency of SSD under varying batch sizes and hardware settings? This would clarify whether the observed TPS gains translate to practical speed improvements.

3. Do the authors envision the self-speculative mechanism generalizing to other generation paradigms—such as masked autoregressive or multimodal diffusion models—and if so, what architectural adjustments would be required?

---

### Official Review · Reviewer_wB8H · 2025-10-31

**Soundness:** 3
**Presentation:** 3
**Contribution:** 3
**Rating:** 4
**Confidence:** 4

**Summary:**

This paper introduces a method to accelerate dLLMs by applying the idea of Speculative Decoding.

Specifically, during each generation step, the dLLM first performs a forward pass. In addition to generating the next token as usual, it simultaneously produces several draft tokens. The dLLM then leverages batching to verify these draft tokens in parallel and accepts a subset of them, thereby boosting generation efficiency. The authors conducted experiments on the LLaDA and Dream series of dLLMs and demonstrated the method's effectiveness across established benchmarks including GSM8K, MATH, HumanEval, and MBPP.

**Strengths:**

1. The central idea of this paper is sound and reasonable, and its effectiveness has been comprehensively validated through experiments.
2. The paper includes a performance bottleneck analysis and discusses the suitability of SSD, which is meaningful.

**Weaknesses:**

1. **Regarding the trade-off between parallel verification efficiency and losslessness.** In the traditional auto-regressive model's SD mechanism, the target model utilizes tree attention to perform parallel verification. In contrast, the SSD proposed in this paper adopts the method of batching to perform parallel verification, which significantly increases the cost of verification. I understand that adopting batching is to ensure the losslessness of inference in dLLM, due to the characteristic of bidirectional attention. However, I still believe that the batching mechanism constitutes a great limitation to SSD achieving a higher speedup ratio. Furthermore, considering that SSD has already adopted the dual-cache mechanism, and this mechanism is theoretically already a lossy optimization, thus insisting on strict losslessness in the SD part seems to be of little significance.
2. **Limited Scope Regarding Sampling.** The current paper appears to focus exclusively on the greedy decoding scenario. It fails to address how SSD can effectively accelerate the generation process when outputs are produced via sampling methods.

**Questions:**

Please see the weakness.

---

### Note · Authors · 2025-11-18

I have read and agree with the venue's withdrawal policy on behalf of myself and my co-authors.